# Examining the Efficacy and Safety of Combined Locoregional Therapy and Immunotherapy in Treating Hepatocellular Carcinoma

**DOI:** 10.3390/biomedicines12071432

**Published:** 2024-06-27

**Authors:** Nojan Bajestani, Gavin Wu, Ahmed Hussein, Mina S. Makary

**Affiliations:** 1College of Medicine, The Ohio State University, Columbus, OH 43210, USA; gavin.wu@osumc.edu (G.W.); ahmed.hussein@osumc.edu (A.H.); 2Division of Vascular and Interventional Radiology, Department of Radiology, The Ohio State University Wexner Medical Center, 395 W 12th Avenue, Columbus, OH 43210, USA; mina.makary@osumc.edu

**Keywords:** hepatocellular carcinoma, locoregional therapy, immunotherapy, combination therapy, progression-free survival, overall survival

## Abstract

More than 800,000 people worldwide are diagnosed with HCC (hepatocellular carcinoma) each year, with approximately 700,000 deaths alone occurring in that same year. Treatment of HCC presents complex therapeutic challenges, particularly in intermediate and advanced stages. LRTs such as transarterial chemoembolization (TACE) and ablations have been the mainstay treatment for early to intermediate-stage HCC, and systemic therapies are used to treat intermediate-late-stage HCC. However, novel literature describing combining LRT with systemic therapies has shown promising results. This review explores recent advances in both liver-directed techniques for hepatocellular carcinoma, including bland transarterial embolization, chemoembolization, radioembolization, and ablative therapies in conjunction as well as with systemic therapies, with a focus on combination therapies, patient selection, procedural technique, periprocedural management, and outcomes. Our findings suggest that LRT combined with systemic therapies is a viable strategy for improving progression-free survival and time to progression for patients with intermediate-to-late-stage HCC. However, further investigation is required to refine treatment protocols and define patient cohorts that would benefit the most.

## 1. Introduction

Despite numerous advances in treatment during the last decade, hepatocellular carcinoma (HCC) remains a leading cause of cancer-related deaths globally, making it a considerable public health concern [1,2,3]. The treatment and prognosis of HCC vary significantly based on the stage of diagnosis. Numerous staging systems are used to predict outcomes for HCC patients; these outcomes examine overall survival, progression-free survival, and adverse events in HCC following intra-arterial treatments [4]. These include the Okuda system, the Cancer of the Liver Italian Program (CLIP) score, and the widely used Barcelona Clinic Liver Cancer (BCLC) classification scheme [2,4]. The modified Barcelona Clinic liver criteria (BLCL) categorize HCC from very early stages (Stage 0) to terminal stages (Stage D), guiding the selection of therapeutic approaches such as ablation, resection, and systemic therapies according to tumor characteristics and liver function [4,5]. Another commonly used classification system is the Child–Pugh criteria, which, although it is not a staging system itself, is frequently used in conjunction with HCC staging systems [5]. These criteria assess the severity of liver disease and can determine prognosis and treatment strategies for HCC patients using factors such as bilirubin levels, albumin levels, prothrombin time, ascites, and hepatic encephalopathy to classify liver function into three classes: A, B, and C [1,5]. Early-stage HCC, which is defined using the BLCL as 3 or fewer lesions less than or equal to 3 cm in diameter with preserved liver function, generally has a favorable prognosis. Curative treatments such as surgical resection, liver transplantation, or local ablative therapies are mainstay therapies [2,3,5]. The intermediate stage (BCLC B) is defined as a multinodular disease that is unresectable with preserved liver function [5]. These patients are generally treated with transplants or locoregional therapies, the two most common being transarterial chemoembolization (TACE) and transarterial radioembolization (TARE) [3,5]. However, these patients may become candidates for systemic therapies under certain conditions (Figure 1) [3,5,6].

As compared to early-intermediate-stage disease, advanced and terminal-stage HCC (BLCL C and D) is more heterogeneous in its tumor burden, distribution, and underlying liver function, which significantly impacts the efficacy of locoregional therapies (LRT) [6,7]. For these patients, systemic therapies such as Sorafenib or immunotherapies like Nivlumab offer benefits in survival yet come with side effects such as hand–foot syndrome, hypertension, and gastrointestinal issues that limit the tolerability of the treatment [7,8,9]. Moreover, even if patients can undergo curative resection, 70% suffer from recurrence within 5 years [10]. To improve survival for intermediate or advanced HCC patients, current trials are investigating the risks and benefits of combining LRT and systemic therapy such as sorafenib or ipilimumab. These combination therapies can allow patients to start effective therapies prior to liver decompensation, minimize the inherent side effects of systemic therapies, and potentially raise both cure rates and progression-free survival [9,11]. This review examines recent advancements in liver-directed interventions for HCC, such as bland transarterial embolization, chemoembolization, radioembolization, and ablative treatments, alongside the integration of systemic therapies. It focuses on the utilization of therapies, procedural techniques, periprocedural management, the criteria for patient selection, and treatment outcomes. 

## 2. Methods

A broad literature search was conducted across multiple electronic databases, including PubMed, Scopus, and the Web of Science. This search encompassed studies published from January 1992 to April 2024. Keywords used in the search included “hepatocellular carcinoma”, “locoregional therapy”, “immunotherapy”, “transarterial chemoembolization”, “radioembolization”, “ablation”, “systemic therapy”, and “combination therapy”. In addition, reference lists of relevant articles and reviews were manually examined to identify any further studies not captured in the initial database search.

The inclusion criteria for this review were studies focusing on HCC and investigating the use of LRTs such as TACE, radioembolization, or ablation in combination with systemic therapies, including immunotherapies. Only studies published in peer-reviewed journals and available in English were considered. Exclusion criteria included studies that did not specifically address HCC, studies evaluating only monotherapies without combination approaches, case reports, editorials, opinion pieces, and non-English publications.

Between three independent reviewers, 160 articles were identified, and data was extracted. Data extraction involved summarizing key information from the selected studies, including study characteristics (author, year of publication, study design, sample size), patient characteristics (age, sex, liver function, HCC stage), details of the locoregional and systemic therapies used, outcomes measured (progression-free survival, overall survival, time to progression, response rates, adverse events), and key findings and conclusions. 

## 3. Results

### 3.1. Ablation

#### 3.1.1. Technique and Periprocedural Management

Ablative therapies are a cornerstone in the management of HCC, particularly for early-stage patients with preserved liver function (Child–Pugh A/B or BLCL Stage 0 and A) who are unsuitable for surgery [11]. The most common ablation techniques for the treatment of HCC include radiofrequency ablation (RFA), microwave ablation (MWA), cryoablation, laser-induced interstitial thermotherapy (LITT), and high-intensity focused ultrasound (HIFU) [12]. The selection of an appropriate ablative technology often depends on the specific expertise available at a treatment center and historical preferences. RFA has traditionally been the most used method [13,14]. However, MWA has recently become more popular as an alternative due to its effectiveness. Cryoablation was frequently employed in earlier treatments but has seen a decline in use because of significant complications such as cryogenic shock, acute renal failure from myoglobinuria, and coagulopathy [15,16].

In thermal ablative therapies such as RFA or MWA, a needle is inserted percutaneously into the liver, usually with the guidance of ultrasonography (US) or computed tomography (CT). Once inserted, radiofrequency ablation uses high-frequency electrical currents to induce liquefactive necrosis of the tumor [11,12]. The ablation zone includes the area originally occupied by the tumor, extending to include a margin of 5–10 mm of the surrounding liver tissue that is also treated [12]. On the other hand, MWA uses an electrode to emit microwaves and trigger necrosis [13]. MWA has the capability to treat multiple tumor sites at once, rapidly reach effective temperatures, produce larger and more clearly defined ablation zones, and is less affected by the cooling effects of nearby blood vessels [17]. In contrast, RFA typically creates smaller and less uniformly shaped ablation zones and is more susceptible to these cooling effects. However, RFA provides the benefit of minimizing energy transfer to critical structures such as bile ducts or large blood vessels, enhancing its safety profile in certain clinical scenarios [13,14,17]. In cryoablation, compressed, cold gases are channeled via hollow probes directly into the tumor under imaging guidance; the tissue is then frozen and destroyed [18]. Indications for cryoablation, as well as survival rates, are comparable to those of RFA [19,20].

Recent advancements have included the use of contrast-enhanced ultrasound (CEUS) and innovative fused imaging technologies to better delineate treatment areas [20]. The National Comprehensive Cancer Network (NCCN) recommends routine follow-up imaging using CT or MRI every 3–6 months for the first two years, alongside monitoring of serum alpha-fetoprotein (AFP) levels [21]. Complications of thermal ablation include major vascular complications, which occur in up to 3.1% of cases and include high-risk pathologies such as bleeding and pseudoaneurysms, among others [22,23]. Other complications involve damage to biliary structures and post-ablation syndrome, where patients experience malaise, fevers, and chills within the first week [24].

#### 3.1.2. Patient Selection

Ablation is preferred over surgical resection or transplantation for early-stage patients (BCLC 0 and A) or those unsuitable for surgery due to significant comorbidities, portal hypertension, poor hepatic function, or intolerance to general anesthesia [24,25]. Specifically, the patients that derive the most significant benefits are those with a single tumor less than 5 cm in diameter or up to three tumors less than 3 cm in diameter, absence of portal vein thrombosis, Child–Pugh Class A or B, and no significant coagulopathy [18,24]. Ablation is a viable, potentially curative alternative for these early-stage patients, with some studies showing survival rates comparable to surgical resection despite poorer initial liver function [25,26]. RFA is not recommended for tumors that are not near central biliary structures, the gallbladder, stomach, or subdiaphragmatic [26]. However, techniques such as hydro-dissection, which involves injecting a 5% dextrose solution to create a barrier between the tumor and sensitive areas, can mitigate the risk of damage during thermal ablation [27,28]. Although not curative, combining ablation with TACE has favorable outcomes and has been known to be used to downstage patients or as a bridge to transplant [29].

#### 3.1.3. Prognostic Factors and Outcomes

When comparing surgical resection and ablation as treatment options for early-stage patients (BCLC 0 and A), surgical resection has generally been considered to have superior 1-, 3-, and 5-year overall survival and disease-free survival rates than RFA [24,26]. However, RFA can be an alternative option for patients with tumors < 3 cm that are not suitable for resection [24]. One meta-analysis noted significantly better hepatic function one week post-treatment, fewer post-operative complications, and shorter hospital stays for RFA patients as compared to surgery [24,25]. Similarly, another study by Shina et al. found that RFA can be locally curative and is a safer option than resection, with a median 5-year survival rate of 60% [11,18,24]. Studies have shown that percutaneous cryoablation and percutaneous RFA have comparable long-term therapeutic outcomes, with recurrence-free survival rates and overall survival rates [18,21]. One retrospective review noted an average progression-free survival of 24 months with functional outcomes, as defined by Eastern Cooperative Oncology Group (ECOG) scores, maintained or improved in 91.4% and 83.3% of patients at 6 months and 12 months, respectively [30]. Recent cryoablation technologies have certain advantages over RFA, including improved visualization, improved pain, and a lower risk of severe damage to biliary and vascular structures [11,18,26,31]. Wang et al. compared cryoablation and RFA in patients with one or two HCCs ≤ 4 cm in diameter and noted that the local tumor progression rate was significantly lower in the cryoablation group compared to the RFA group (5.6% versus 10%) [32]. This could be due to the larger ablation zone created relative to RFA. However, for lesions > 3 cm, the difference in local tumor progression rate increased (7.7% versus 18.2%) in the cryoablation group [32].

Multiple studies have compared the benefits of RFA when used in combination with TACE. RFA reduces overall cellular resistance, enhancing the effectiveness of chemotherapy delivered via TACE. This process allows higher concentrations of chemotherapy to accumulate near the tumor’s vascular bed, at the edges of the ablated area [33,34,35,36]. When TACE precedes RFA, it preferentially targets and destroys tumor cells located at the periphery, reducing vascular heat-sink effects during subsequent RFA. This sequence is particularly effective for larger lesions over 3 cm, resulting in more thorough central necrosis when RFA is performed within four weeks after TACE [33,36]. For patients with intermediate-to-advanced stage HCC (BCLC B/C), ablation can down-stage patients to transplantation [24,37]. However, one retrospective study noted no notable differences in overall survival between patients who underwent ablation to meet the Milan criteria prior to transplantation and those who were transplanted while already meeting the criteria [38].

Future work should evaluate the risks and benefits of newer ablation technologies compared to one another and surgical resection for patients with early-intermediate-stage HCC. Similarly, there is a gap within the literature on the optimal imaging technologies used to guide ablative therapies for HCC treatment.

## 4. Transarterial Embolization

### 4.1. Technique and Periprocedural Management

Intraarterial therapies operate on the principle that hepatocellular tumors greater than 2 cm primarily derive their blood supply from hepatic artery branches, while the normal liver parenchyma receives its main blood supply from the portal vein [33,39,40]. Transarterial Embolization (TAE) is a widely used locoregional therapy in the treatment of unresectable intermediate and advanced HCCs. TAE operates by obstructing hepatic artery blood flow to the tumor, inducing ischemia and cell death while preserving normal hepatic parenchyma [40]. Both particulate and liquid embolic agents are used in this process, with TAE sometimes referred to as “bland” embolization due to the non-chemotherapeutic or radiative nature of the embolic agents [41,42]. Identifying the key hepatic artery branches that supply the tumor is essential for optimizing its efficacy and minimizing damage to non-targeted liver tissue [40,42,43,44]. Treatment typically involves administering embolic agents, often microparticles sized between 40 and 120 μm, that occlude the tumor’s arterial supply [12,41,45,46]. Similarly, the most commonly used embolic agents have historically ranged from gel foam to polyvinyl alcohol [40]. The treatment strategy can vary from lobar treatment for multifocal disease to precise segmental treatment for isolated tumors [44,47,48].

Arterial access is usually obtained from either the femoral, radial, or brachial artery and is usually conducted under moderate sedation, though general anesthesia may be necessary for some individuals [47,48]. A selective angiographic catheter is used to access the celiac artery. A microcatheter is then used to navigate to the hepatic arterial branch supplying the tumor, with angiography utilized to verify the correct artery. Afterward, microparticles are infused through the arterial catheter until blood flow stasis is achieved. Finally, the arterial puncture site is sealed with manual pressure or a closure device [45].

Prophylactic antibiotics are often given before undergoing TAE. The importance of antibiotics is heightened for patients with altered sphincter function due to the risk of post-procedure infection [40,49,50]. In such cases, pre-procedural antibiotics and bowel preparation might be advantageous. Commonly used antibiotics include moxifloxacin, levofloxacin, ciprofloxacin, or metronidazole, among others. For instance, one retrospective review reported the benefits of administering oral moxifloxacin (400 mg) three days before and 17 days after the procedure to prevent hepatic abscesses in high-risk patients [51]. There have also been benefits noted with administering a combination of levofloxacin and metronidazole two days prior to the procedure and continuing the medications for two weeks post-procedure alongside a bowel regimen of neomycin and erythromycin [51,52]. Similarly, there have been benefits noted for using prophylactic IV antibiotics for high-risk patients. Nevertheless, the choice of prophylactic antibiotics depends on local preference, with most institutions opting for agents with extensive Gram-negative coverage [45]. Additional standard pre-procedural preparation includes ensuring proper hydration, antiemetics, antihistamines, and steroids such as dexamethasone and hydrocortisone [40,52]. The use of post-procedural antibiotics remains debated and should be considered on a case-by-case basis, especially for patients with a history of biliary issues or interventions [41,52]. Proper anticoagulation management per existing guidelines is also crucial [53,54]. Follow-up evaluations, including imaging and lab tests scheduled 4–6 weeks after the procedure and periodically every 3–6 months, help monitor the treatment’s success and the disease’s progression [40,46]. Advanced imaging techniques like angiography or cone-beam CT are used throughout the procedure to ensure precise targeting and minimize risks [55].

### 4.2. Patient Selection

TAE is typically reserved for patients who are not candidates for surgery but have a liver-dominant disease. The selection process for TAE involves a comprehensive evaluation of clinical and serologic assessments, functional status, and various scoring systems such as Albumin-Bilirubin (ALBI), Child–Pugh, Model for End-stage Liver Disease (MELD), and ECOG performance status [56,57,58,59]. Although those in BCLC class C can derive benefits, research shows that TAE is particularly beneficial for patients classified within BCLC class B [57]. In cases of patients classified as BCLC class A, TAE can maintain eligibility for liver transplantation [58]. Prior to treatment, diffusion-weighted imaging (DWI) can be used as a biomarker to monitor and predict tumor responses to LRTs [60].

TAE is contraindicated for patients with conditions such as decompensated cirrhosis (Child–Pugh B8 or higher), compromised portal venous flow, creatinine clearance less than 30 mL/min, a high tumor burden, severe comorbidities, untreated esophageal varices, and elevated liver function markers [61,62,63]. In the case of cirrhotic liver disease, patients are more likely to experience complications from decompensated cirrhosis than from tumor-related events [45]. Similarly, biliary obstruction is a contraindication to TAE; however, for these patients, biliary decompression can be performed with retrograde stenting or percutaneous drainage [45]. 

### 4.3. Prognostic Factors and Outcomes

Research investigating the advantages of TAE over TACE and TARE is ongoing and yields mixed results. While locoregional therapies typically show a survival advantage over best supportive care, particularly for TACE, the efficacy of TACE compared to TAE remains similar in terms of overall survival, based on several studies [64,65,66,67]. However, it has been noted that patients treated with TACE were more likely to experience adverse events secondary to chemotherapy as compared to TAE [64,68]. Notably, Kluger et al. reported that TAE patients had a lower likelihood of requiring retreatment before transplantation compared to those undergoing TACE [65]. Moreover, Malagaris et al. conducted a multicenter randomized controlled trial comparing DEB-TACE with TAE, which showed an improvement in time to progression with DEB-TACE but not in overall survival [67]. Given that induced ischemia from embolotherapy is a key mechanism in tumor destruction and bland embolization avoids the costs and side effects of chemotherapy, TAE remains a viable option for appropriately selected patients [69]. Unfortunately, recurrence is common for patients treated with TAE, both within the treated region and at other sites within the liver [45].

The most common risk with TAE is postembolization syndrome (PES), a self-limited inflammatory response with characteristic symptoms including right upper quadrant (RUQ) pain, fever, nausea, vomiting, and transaminitis. PES can manifest within the first few days following the procedure [40,41]. However, both its severity and duration are linked to the extent of ischemia in healthy tissue and the patient’s liver function [41]. Other major complications include hepatic decompensation, renal and biliary injuries, abscesses, and pulmonary embolization [46].

## 5. TACE

### 5.1. Technique and Periprocedural Management

TACE involves administering chemotherapy through a catheter placed in the hepatic artery that supplies the tumor [24]. This approach leverages the ability to both localize chemotherapy to tumor sites and restrict blood supply without surgical intervention. The chemotherapy can be delivered as drug-eluting beads (DEB-TACE) or as emulsions with iodized oil or cytotoxic drugs followed by embolic agents (cTACE) [70,71]. Approaches using DEB-TACE provide better standardization and can minimize hepatotoxicity [70,72]. Similarly, DEB-TACE enables a steady administration of therapeutic agents, which can prolong local exposure at the tumor site and minimize potential systemic effects [73]. The most commonly used chemotherapeutic agents in c-TACE are 5-fluorouracil, doxorubicin, mitomycin C, and cisplatin [46,74]. 

There are multiple technical approaches involving TACE: selective TACE, super-selective TACE, and ultra-selective TACE. Selective TACE, targeting a segmental artery, and super-selective TACE, targeting the distal part of a sub-segmental hepatic artery, both enhance survival compared to broader lobar administration [46,71]. Ultra-selective TACE involves the precise delivery of a lipoidal agent to the hepatic artery until the tumor’s portal venous supply becomes visible, which helps to ensure complete tumor ischemia and has been shown to enhance local tumor response in multiple studies [24,46,63,71]. Periprocedural management for TACE is identical to TAE (Section 3.1).

### 5.2. Patient Selection

According to the BCLC classification, TACE is the first-line treatment for intermediate-stage HCC, including unresectable cancer that cannot be ablated and has not spread to major blood vessels [24,63,71]. The BCLC system also incorporates the concept of treatment migration—TACE should be used in early-stage HCC patients whose recommended treatments are not feasible or have failed [24]. Furthermore, eligible patients must have preserved liver function (Child–Pugh B) and no evidence of portal vein thrombosis [24,70]. Additional uses of TACE involve its use as bridging therapy for transplants and for tumors larger than 5 cm, in which multiple treatments are required [63,75]. 

TACE has a low overall complication rate of under 5%. Major complications include liver decompensation with ascites (2.8%), acute cholecystitis (1.5%), pancreatitis (0.9%), and renal failure (0.6%). Postembolization syndrome appeared in 20% of patients and presented with fever, abdominal pain, nausea, vomiting, and elevated transaminase levels immediately or up to ten days post-procedure [24,76]. 

### 5.3. Prognosis and Outcomes

Survival benefits linked with TACE are significant; selective and super-selective TACE have achieved complete response rates of approximately 40–50% and 5-year survival rates of 20–35% [24,69,75,77]. On the other hand, some articles note response rates as high as 86.6% [78]. However, it is essential to note that TACE is predominantly studied in the intermediate to advanced stages of HCC, not the early stages. A small prospective study showed a 3-year survival rate of 80% for selective TACE in early HCC (BCLC 0 or A) [24,79]. 

TACE is most beneficial for patients with fewer lesions and good liver function (BCLC B) [24,79]. Innovations such as balloon-occlusion TACE (B-TACE) and microvalve infusion catheters have improved tumor targeting and increased rates of tumor necrosis, though they have not yet shown clinical outcome improvements [29]. In particular, B-TACE has demonstrated higher complete response rates compared to conventional TACE. The technique is also well-established for bridging and downstaging to transplantation [29,79]. 

In a palliative setting, employing conventional TACE has been shown to extend patient survival by 8–11 months compared to just providing the best supportive care. However, this benefit depends on accounting for all known contraindications and additional factors related to the tumor and liver disease that could negatively affect the prognosis. While drug-eluting TACE has been more effective in achieving localized responses, it has not yet demonstrated a survival advantage over conventional TACE [24]. Additionally, an improved prognosis after liver transplantation is linked to positive responses to neoadjuvant local treatments [24,75]. 

Recent innovations, such as B-TACE and microvalve infusion catheters, have further refined TACE by improving tumor targeting and increasing rates of tumor necrosis [24,29,46]. Despite these technical advancements, challenges persist in enhancing TACE outcomes, including the absence of technical standards and the limited effectiveness of specific chemotherapeutic agents. Future advancements in curative TACE will depend on the development of these standards and the formulation of more effective and tolerable chemotherapeutics.

## 6. TARE

### 6.1. Technique and Periprocedural Management

For patients receiving treatment for intermediate-stage HCC, TACE has the highest quality of evidence among LRTs [2,80,81]. An alternative therapy, transarterial radioembolization, also known as selective internal radiotherapy (SIRT), is a newer technology that involves the delivery of Yttrium-90 (Y90)-coated microspheres into the hepatic artery using a catheter [81]. Prior to the procedure, preprocedural angiographic mapping is conducted for 1–2 weeks to identify anatomic abnormalities [82]. Technetium-99m labeled macroaggregated albumin (99mTc-MAA) is used with single-photon emission computed tomography (SPECT) to determine the hepatopulmonary fraction; high fractions may increase the likelihood of radiation pneumonitis after TARE [40,83,84]. 

The actual TARE procedure is performed similarly to other locoregional endovascular approaches, targeting the tumoral disease in a lobar or segmental fashion. For further information on this, see TAE Technique, Section 4. The effects of treatment are observed much later than with TACE or TAE, typically 12 weeks after the procedure. At this time, follow-up imaging and labs are conducted to identify any adverse effects of treatment complications [42,85].

### 6.2. Patient Selection

TARE has similar indications and contraindications to TAE and TACE—a total bilirubin level of up to 2 mg/dL is acceptable, while a history of prior liver radiation or signs of liver decompensation, such as encephalopathy, is not [85]. Another significant contraindication is >20% hepatopulmonary shunting due to increased radiation exposure [82]. TARE has unique applicability in patients with portal vein thrombosis due to its reduced embolic effect [41]. Similarly, several studies have demonstrated the safety of TACE in cases where tumors have infiltrated either a main or lobar portal vein branch [85]. 

Currently, TARE is recommended after TACE as first-line therapy for BCLC Class B patients [58]. However, expert recommendations from AASLD and NCCN do not consider radioembolization inferior for unresectable intermediate-stage HCC patients [86]. For BCLC 0 and A patients, radiation segmentectomy with intraarterial 90Y-SIRT is safe and effective [87,88]. Neoadjuvant radiation lobectomy is also a viable option to increase the function of the contralateral future liver remnant in patients planning to undergo resection, avoiding the risks associated with portal vein embolization [85,89]. Additionally, TARE can be used to maintain or encourage transplant/resection eligibility through bridging and enhance overall survival in BCLC C patients [90,91]. 

### 6.3. Prognostic Factors and Outcomes

In multiple cohort studies, TARE has shown efficacy in the treatment of unifocal HCC [82,92,93]. Although TARE has not been adopted as a primary treatment in guidelines due to a lack of robust, controlled data, it has recently been incorporated into the BCLC guidelines as a second-line therapy for early-stage HCC [80,93]. A single-center randomized controlled trial compared overall survival for patients with intermediate-stage HCC who received either TACE or TARE and found no difference in overall survival (OS). Patients enrolled in this study were BCLC A-C. This study also noted a longer time to progression (TTP) for patients receiving TARE as compared to TACE [80,93,94]. 

Prognosis after TARE is most associated with baseline patient stage (BCLC, Child–Pugh), ECOG, tumor burden, and extrahepatic disease [40]. According to a 2016 meta-analysis by Lobo et al., overall survival and complication rates for TARE are similar to those of TACE, but the prospective trial PREMIERE demonstrated a longer TTP for TARE [95]. Another randomized trial showed higher quality of life scores for TARE patients compared to TACE [96]. Salem et al. reported excellent outcomes in a prospective study, including an overall survival of 47.3 months for Child–Pugh A patients and 27 months for Child–Pugh B patients [97]. With more contemporary approaches such as radiation segmentectomy, response rates, tumor control, and survival outcomes have been comparable to curative-intent treatments (e.g., resection, transplantation, ablation) at 5 years [97,98]. 

Recent clinical trials and studies, specifically the DOSISPHERE-01 and TARGET trials, highlighted the benefits of personalized dosimetry, showing improved response and survival rates in patients with advanced disease stages [99]. Additionally, two studies following the LEGACY study and the updated BCLC treatment algorithm reported excellent outcomes for early- and intermediate-stage HCC [100]. The RASER study indicated an 83% complete response rate with a mean duration of response of 516 days in early-stage BCLC-A patients [101]. The TRACE trial showed significant survival improvements in patients with early- or intermediate-stage disease undergoing radioembolization compared to drug-eluting bead chemoembolization [93,102]. Similarly, functional outcomes can be improved in a high percentage of populations. Makary et al. conducted a retrospective study and noted that functional outcomes (Eastern Cooperative Oncology Group) were maintained or improved in 79.6% and 76.1% of patients by 6 months and 1 year post-treatment, respectively [103].

While TARE is an effective treatment, it is not without potential complications. Adverse events arise mainly from unintended effects on non-cancerous tissue or issues related to catheter insertion and manipulation [80,104] (Lo CM, Sacco). Complications include the following: liver failure or radio-induced liver disease, with an incidence of up to 4%; biliary complications in less than 10% of cases; post-radioembolization syndrome affecting 20–55% of patients; gastrointestinal issues in less than 5%; and radio-induced pneumonia in less than 1% [104].

## 7. Systemic Therapies/Immunotherapies

### 7.1. Technique and Periprocedural Management

Unfortunately, more than 50% of HCC is discovered at an advanced stage (BCLC D), when systemic therapy is the main and only treatment available. When liver-directed treatments, such as LRT, resection, or transplant, are not an option, systemic therapy is chosen. These therapies include tyrosine kinase inhibitors (TKI), monoclonal antibodies, and immune-checkpoint inhibitors. There are multiple trials exploring the benefits of each of these systemic therapies alone, combined with each other, or in conjunction with LRTs. However, of these therapies, sorafenib stands out as a first-line systemic therapy and has been noted to prolong survival and progression-free survival in advanced-stage patients [105]. It has been the standard first-line therapy for approximately 10 years. Sorafenib works as a multi-kinase inhibitor that inhibits tumor angiogenesis and blocks the expression of the vascular endothelial growth factor receptor (VEGFR). Lenvatinib is another first-line therapy TKI that can be administered over sorafenib for individuals who have failed or experienced adverse reactions to other medications. In contrast to sorafenib, lenvatinib inhibits VEGFR, FGFR, and PDGFR, providing a broader mechanism of action that also impacts fibroblast growth factor pathways involved in tumor growth. Additional options include cabozantinib, nivolumab, and pembrolizumab, among others [106]. Of these, nivolumab and pembrolizumab have shown the most promising results for progression-free survival, though mainly in phase II studies [107]. Nivolumab and pembrolizumab are immune checkpoint inhibitors that target the programmed cell death protein 1 (PD-1) pathway. PD-1 is expressed on the surface of most immune cells and, when bound to its ligands, PD-L1 and PD-L2, causes an inhibitory effect, allowing the tumor cells to evade host defenses. Similarly, CTLA-4 is also expressed on various immune cells, including T-cells, and also produces a suppressive effect on the immune system when activated by its ligand [108]. TKIs are administered orally, and ICIs are generally delivered via injection or in pill form.

### 7.2. Patient Selection

Current guidelines include systemic therapies for treating patients with intermediate- to advanced-stage HCC. This includes patients with intermediate-stage HCC (BCLC Class B) characterized by diffuse, infiltrative bilobar involvement and those in advanced stages (BCLC C and D) who exhibit extrahepatic invasion or end-stage liver function and are not eligible for curative treatments due to the advanced nature of their disease or the presence of comorbid conditions [4]. First-line systemic therapies, usually TKIs, are also considered for certain BCLC B patients who have failed LRTs like ablation and TACE. Furthermore, patients with good performance status (ECOG 0-1) and preserved liver function (Child–Pugh A) are generally considered suitable candidates [7]. Molecular profiling and biomarkers have also increasingly been used to refine patient selection and tailor systemic therapy, ensuring patients receive the most effective treatment based on their tumor’s specific characteristics, such as levels of AFP, CTNNB1 mutations, and glypican-3 (GPC3) mutations [109].

### 7.3. Prognostic Factors and Outcomes

The efficacy of Sorafenib as a first-line treatment in those with unresectable HCC was evaluated in two phase III trials: the European Sorafenib Hepatocellular Carcinoma Assessment Randomized Protocol (SHARP) research and the Asia–Pacific trial. These studies included patients with Child–Pugh Class A cirrhosis, no prior systemic therapy, and a life expectancy of at least 12 weeks [106]. The SHARP study demonstrated a median OS of 10.7 months in the sorafenib group compared to 7.9 months in the placebo group (*p* < 0.001) [110]. In the Asia–Pacific trial, the median OS was 6.5 months for the sorafenib group and 4.2 months for the placebo group. (*p* = 0.014) [111]. However, both the SHARP and Asia–Pacific trials showed low response rates of 2% and 3.3%, respectively, to sorafenib [106,110].

Further evaluation of Sorafenib’s efficacy was examined in two large observational studies, GIDEON and INSIGHT. The Global Investigation of Therapeutic Decisions in Hepatocellular Carcinoma and of its Treatment with Sorafenib (GIDEON) study assessed the tolerability of sorafenib in advanced HCC in different Child–Pugh classes, revealing a comparable incidence of adverse effects that lead to discontinuation amongst Child–Pugh classes A and B (17% and 21%, respectively). Common adverse effects included hand–foot skin reactions, diarrhea, and exhaustion. The GIDEON study also reported a median overall survival of 13.6 in Child–Pugh A patients compared to 5.2 in Child–Pugh B patients [112]. The INSIGHT trial, which also examined safety and efficacy, demonstrated a median overall survival of 17.6 months in patients in Child–Pugh Class A, 8.1 months in patients in Child–Pugh Class B, and 5.6 months in patients in Child–Pugh Class C [113]. In the Phase III study REFLECT, sorafenib was compared to lenvatinib, another tyrosine kinase inhibitor (TKI), demonstrating that the median OS was 13.6 compared to 12.3 months with sorafenib. The main side effects associated with TKIs are outlined in Table 1. The most common side effects include hand–foot skin reactions, hypertension, diarrhea, and fatigue. These side effects are often managed with supportive care, dose adjustments, or temporary discontinuation of therapy to ensure patient comfort and adherence to treatment [114]. Lastly, the development of biomarkers to predict response to TKI therapy is an area of active research. Prognostic factors such as vascular invasion, elevated AFP levels, and an elevated neutrophil-to-lymphocyte ratio were associated with reduced survival [115].

Additionally, several other systemic drugs have been studied as potential monotherapies, including the multikinase inhibitors regorafenib and cabozantinib, the VEGF inhibitor ramucirumab, and the anti-PD-1 pembrolizumab. Regorafenib and cabozantinib have been shown to have a statistically significant increase in overall survival compared to placebo in those with disease progression on sorafenib [116,117]. The REACH trial explored the effects of ramucirumab in patients with advanced HCC who had previously received sorafenib. Although the overall survival did not significantly increase in the REACH trial, the REACH-2 demonstrated better survival in those with an AFP of 400 ng/mL or more taking ramucirumab compared to placebo [118,119]. Pembrolizumab was also investigated in the phase III Keynote-240 trial as a therapy in patients with advanced HCC and previous treatment with sorafenib. This study showed a median OS of 13.9 months for pembrolizumab versus 10.6 months for placebo [120]. 

Several studies have evaluated the efficacy of combined systemic therapies in patients with unresectable hepatocellular carcinoma. In a phase 3 study, COSMIC-312, the combination of cabozantinib and atezolizumab, an anti-PD-L1 antibody and VEGF inhibitor, was compared to sorafenib as a first-line treatment. The median OS difference between the two groups was not statistically significant [121]. However, another study showed that the median OS was 5.8 months longer for the combination of atezolizumab and bevacizumab compared to sorafenib [122]. Additionally, the phase III HIMALAYA trial compared different regimens of tremelimumab (CTLA-4 antibody) and durvalumab (anti-PD-L1) to both durvalumab alone and sorafenib alone. The study yielded promising results, as the median OS in a tremelimumab and durvalumab arm was 16.4 months, compared to 13.8 months in the sorafenib arm. Additionally, the objective response rate (ORR) was 20.1% with the tremelimumab and durvalumab regimens, 17.0% with the durvalumab arm, and 5.1% with sorafenib, respectively [123].


**COMBINATION THERAPIES**

**(A) TACE/MOLECULAR TARGETED THERAPIES**


## 8. TACE/Sorafenib

### 8.1. Technique and Periprocedural Management

While LRTs like TACE have been established as standard care for treating intermediate-stage HCC, many patients still encounter complications such as rapid cancer progression or recurrence [124,125]. Combining TACE with systemic and immunotherapy-based treatments, which have traditionally been used for more advanced, unresectable HCC, represents a novel and effective approach [124,125]. This innovative treatment strategy of initiating sequential systemic therapy followed by LRT has shown significant therapeutic advantages, including enhanced progression-free survival (PFS) and OS outcomes [126]. One such combination therapy is sorafenib–TACE. As described in Section 5, sorafenib is a form of molecularly targeted therapy that inhibits tyrosine kinase, suppressing targets such as VEGF, RAF, and platelet-derived growth factor receptor β-pathways (PDGFR) [124,127,128]. This mechanism of disrupting angiogenesis and tumor cell proliferation is central to treating HCC and prolonging patient survival [127,129,130]. As such, multiple clinical trials have investigated combined sorafenib–TACE therapies as a therapeutic option for patients with intermediate-stage HCC, theorizing that the anti-angiogenic effects of sorafenib would normalize tumor vasculature and improve subsequent drug delivery and efficacy with TACE [126,127,131].

The technique of combined sorafenib–TACE therapy depended on each clinical trial, with the most significant variation observed in the sequence of therapy administration. Results from these trials have observed that the sequential application of these treatments could potentially influence treatment efficacy. Cycles of oral sorafenib were administered either before, after, or in conjunction with TACE at a dosage of approximately 400 mg, although specific dosages varied among studies. Additionally, although nearly all trials employed a super-selective approach via the hepatic artery for the general TACE procedure, there were significant variations in the specific types of TACE utilized. For example, the TACE-2 and SPACE trials utilized TACE with drug-eluting beads loaded with 150 mg of doxorubicin, whereas the TACTICS trial employed TACE administration with either epirubicin or miriplatin. Primary differences in technique for each trial can be compared in Table 2 [124,126,127,132,133,134].

Patient monitoring was an important aspect of all trials, with stringent measures in place to ensure patient safety. Baseline screening tests and imaging were conducted, along with laboratory evaluations at various intervals before, during, and after each trial. Laboratory evaluations encompassing hematology, coagulation, biochemistry, and alpha-fetoprotein levels were also conducted at varying intervals during the trial. In response to adverse events, treatment interruptions and reductions in drug dosage were implemented. In many trials, sorafenib dosages were reduced to 400 mg once daily and subsequently to 400 mg every other day. Discontinuation of the drug occurred in cases of disease progression, drug toxicity, patient preference, or as recommended by the investigator. Drug toxicity was assessed using updated versions of the National Cancer Institute Common Terminology Criteria for Adverse Events [124,126,127,132,133,134].

### 8.2. Patient Selection

Many of these trials have selected patients based on several criteria, including overall functional status, liver function, and clinical indices such as Child–Pugh classification, ECOG performance status, and BCLC staging. Inclusion criteria for almost all of these studies included patients aged 18 years or older with unresectable HCC, Child–Pugh A or B cirrhosis, and ECOG performance statuses between 0 and 2. Further details on patient selection for these trials, including exclusion criteria, can be viewed in Table 2 [124,126,127,132,133,134].

### 8.3. Prognosis and Outcomes

Prognostic factors impacting survival for patients undergoing combined sorafenib–TACE therapy for HCC include established indicators such as BCLC stage, Child–Pugh class, and ECOG performance status score, while other metrics include maximum tumor size, tumor number, alpha-fetoprotein concentrations, metastasis/vascular invasion, extrahepatic spread, and sorafenib treatment duration [135,136].

The outcomes for combined sorafenib–TACE therapy in previous studies have yielded inconsistent results. The primary endpoint across the TACTICS trial, SPACE trial, TACE-2 trial, and Post-TACE trial was PFS or TTP. However, only the TACTICS trial exhibited positive results. Regarding OS, the SPACE, TACE-2, and Post-TACE trials, where OS served as the secondary endpoint, did not reveal a significant extension of OS compared to patients treated solely with TACE. Kudo et al. surmised several possible reasons for trial failure. Since the combination of sorafenib–TACE represented a novel therapeutic approach, there was no consensus on a standardized treatment protocol or treatment administration to achieve the optimal therapeutic effect. For instance, the POST-TACE study was conducted prior to the positive results of the SHARP study; as Japanese patients did not participate in either the SHARP or Asia–Pacific studies, physicians in the Post-TACE study—potentially unfamiliar with managing sorafenib’s side effects—administered the drug at low doses and for shorter durations, leading to suboptimal treatment outcomes. As a result, treatment duration was notably reduced in Japanese patients with a sorafenib treatment time of 16 weeks. Similarly, in the SPACE study, treatment duration was 21 weeks, as sorafenib administration was terminated early based on criteria for TACE-untreatable progression [124,126,127,132,133,134].

The TACTICS trial was a landmark investigation within the domain of sorafenib–TACE combination studies, as it demonstrated significant therapeutic benefits for patients receiving combined sorafenib–TACE therapy. In this study, OS and TACE-specific PFS were employed as endpoints instead of the modified Response Evaluation Criteria in Solid Tumors (RECIST). TACE-specific PFS was defined as the interval from randomization to progressive disease (PD) or death from any cause, with PD delineated as untreatable (UnTACEable) progression, denoting the patient’s inability to undergo or derive benefit from further TACE procedures. Utilizing these endpoints, the TACTICS trial demonstrated significant OS prolongation and improved PFS, especially in patients with high tumor burden. Compared to patients receiving TACE alone, the results of the TACTICS trial demonstrated that patients receiving combined therapy had a clinically meaningful median OS (36.2 months with combined therapy vs. 30.8 months with TACE alone, ΔOS = 5.4 months) and a longer updated PFS (22.8 months with combined therapy vs. 13.5 months with TACE alone). These results can be viewed in Table 2, along with the results of earlier trials [126,127].

Adverse events experienced by patients undergoing combined sorafenib–TACE therapy were similar to the side effects experienced by patients undergoing TACE or systemic therapy alone. The most common adverse effects included postembolization syndromes and sorafenib-related toxic effects, such as abdominal pain, fatigue, diarrhea, nausea, and hand–foot skin reactions, among others [124,126,127,132,133,134,136].

The findings from these trials carry significant implications for the advancement of combination therapies in the field. The failures of previous trials prompted an extensive reevaluation of trial design and approach, which has been instrumental in the recent successes of phase 3 clinical trials for HCC treatments. For instance, the definition of “progression” for TACE trials as an endpoint was better defined, which not only improved how TACE was performed in clinical practice but also provided guidance in the development of the successful TACTICs trial. Consequently, the TACTICS trial demonstrated the viability of combining molecular-targeted agents, such as sorafenib, as a feasible treatment strategy in patients with intermediate-stage HCC [126,127].

## 9. TACE/Lenvatinib

### 9.1. Technique and Periprocedural Management

Lenvatinib functions as a small molecule inhibitor of various pathways, including VEGFR, fibroblast growth factor receptor (FGFR), PDGFR, KIT, and RET. Notably, its targeting of the FGF pathway in HCC sets it apart from sorafenib [125]. Similar to sorafenib, its capability to inhibit angiogenesis within tumor proliferation pathways suggests therapeutic potential as a combined therapy with TACE for unresectable HCC. The pairing of lenvatinib and TACE has garnered clinical interest, and several trials have been conducted to investigate its efficacy in the treatment of HCC. Of note, the results from the LAUNCH trial indicate that lenvatinib–TACE (LEN–TACE) combination therapy could offer significant PFS and OS benefits when compared to lenvatinib monotherapy while also providing a similar side effect profile [137].

The LAUNCH trial was a phase III clinical study designed to compare the clinical outcomes of combined lenvatinib–TACE therapy versus lenvatinib monotherapy in patients with advanced HCC. The treatment regimen involved oral administration of lenvatinib at either 12 mg or 8 mg doses, depending on patient weight. Super-selective TACE was then administered one day after lenvatinib initiation and was repeated in cases of incomplete necrosis or tumor regrowth. Based on the investigator’s discretion, the administration of TACE was conducted with either drug-eluting beads preloaded with 75 mg of doxorubicin or an injection emulsion of oxaliplatin or epirubicin with lipiodol. The same technique was then utilized for the patient throughout the study, with lenvatinib continuously administered throughout the study and during TACE [137].

Periprocedural management involves consistent clinical monitoring of patients for side effects and toxicity. Baseline imaging of the celiac and superior mesenteric arteries was performed for all patients to evaluate liver vasculature and circulation prior to treatment. In the event of lenvatinib-related toxicities, dose interruptions were allowed, followed by reductions to 8 mg/day, 4 mg/day, or 4 mg every other day. Adverse events were assessed using the National Cancer Institute Common Terminology Criteria for Adverse Events, Version 4.03. Lenvatinib treatment was discontinued if there was disease progression or unacceptable toxicity during the treatment period [137].

### 9.2. Patient Selection

Similar to sorafenib–TACE combination therapy, LEN–TACE sequential therapy may provide therapeutic benefit when the complete response is not achieved in TACE-unsuitable intermediate-stage HCC [125,137]. The LAUNCH trial accounted for this in its inclusion criteria, selecting patients aged 18–75 years with advanced primary HCC without prior treatment or with initial recurrent advanced HCC following radical resection without postoperative treatment. Inclusion criteria also required at least one measurable liver lesion based on mRECIST; a single intrahepatic tumor (≤10.0 cm) or multiple tumors (≤10 foci) with a tumor burden < 50%; an Eastern Cooperative Oncology Group performance status score of 0 or 1; Child–Pugh Class A; a life expectancy of at least 3 months; and satisfactory blood, liver, and kidney function parameters. Patients with a history of hepatic decompensation, cerebrospinal fluid metastases, other malignancies, or contraindications for TACE were excluded [137].

### 9.3. Prognosis and Outcomes

The prognosis following LEN–TACE treatment was linked to improved survival outcomes in patients with substantial tumor burden, including those with portal vein tumor thrombosis (PVTT), AFP levels of ≥400 ng/mL, three or more intrahepatic tumors, and a primary tumor size of ≥5 cm [137].

The results from the LAUNCH trial indicate that LEN–TACE combination therapy could serve as a viable therapeutic option for patients with intermediate HCC. Compared to lenvatinib monotherapy, the median OS was significantly longer in the LEN–TACE group (17.8 vs. 11.5 months), with a longer median PFS (10.6 months vs. 6.4 months) and a higher response rate according to the modified RECIST assessment (54.1% vs. 25.0%). The improved treatment response demonstrated by LEN–TACE combination therapy may be due to a variety of factors—as both Tanaka et al. and de Stefano et al. have suggested, the administration of TACE therapy serves to debulk intrahepatic tumor burden, which could potentially improve the efficacy of subsequent lenvatinib administration and positively impact patient survival. The positive prognostic factors described above further support this hypothesis, as managing intrahepatic tumor burden is expected to improve liver function, enable prolonged lenvatinib administration, and enhance PFS. Based on these results, LEN–TACE combination treatment could potentially serve as an effective downstaging/conversion therapy, which would allow patients to undergo radical treatment like surgical resection and ultimately improve survival outcomes [137].

LEN–TACE combination therapy was also shown to be safe, with an adverse effect profile similar to that of lenvatinib monotherapy. However, the LAUNCH trial indicated that patients treated with LEN–TACE experienced a significantly higher incidence of abdominal pain, nausea, increased liver enzymes, and hyperbilirubinemia, which might be due to postembolization syndrome as the result of TACE administration. Nonetheless, these adverse events were manageable and did not necessitate dose reductions or interruptions of lenvatinib [137].


**(B) TACE/IMMUNE CHECKPOINT INHIBITORS**


## 10. TACE/Nivolumab

### 10.1. Technique and Periprocedural Management

Given that HCC typically arises in the setting of chronic liver inflammation, it has been hypothesized that HCC may exhibit immunogenic properties and could potentially be targeted by immunotherapeutic agents [130]. Several studies are currently assessing the therapeutic benefit of ICIs in conjunction with TACE in the treatment of unresectable HCC [130].

Nivolumab, an immune checkpoint inhibitor (ICI), inhibits the PD-1 signaling pathway and restores anti-tumor immune activity. It has received approval from the U.S. FDA for use in various cancers, including melanoma, non-small cell lung cancer (NSCLC), and kidney cancer [130]. The IMMUTACE trial, a phase II study, explored the safety and effectiveness of combining nivolumab with TACE. The treatment regimen began with an initial TACE session, followed by the administration of 240 mg of nivolumab within 2 to 3 days. Subsequently, 240 mg of nivolumab was administered every 2 weeks until disease progression, for a maximum treatment period of two years. A second TACE procedure could be performed based on the investigator’s discretion [138]. 

### 10.2. Patient Selection

Combined nivolumab–TACE therapy could offer therapeutic advantages in cases where a complete response is not achieved in intermediate-stage HCC patients deemed unsuitable for TACE. The IMMUTACE trial enrolled patients with histologically confirmed intermediate-stage HCC with limited metastatic disease, an ECOG score of 0 to 2, and a Child–Pugh score of A [138].

### 10.3. Prognosis and Outcomes

While there were no prognostic factors identified with better patient outcomes in the IMMUTACE trial, the study achieved its primary endpoint with an objective response rate of 71.4%, a median PFS of 7.2 months, and an OS of 28.3 months. Treatment was generally tolerable, with only 34.7% of patients experiencing adverse events of Grade ≥ 3. The results of the IMMUTACE trial demonstrate that nivolumab–TACE combination therapy could be a potential therapeutic option for patients with intermediate HCC who had not received prior systemic therapy, and the combination of ICI therapies and TACE remains an emerging field [138]. Indeed, early findings from the phase I trial NCT03143270 indicate that combining ICI therapy with DEB-TACE has antitumor activity and is a safe therapeutic option [130,139]. Another study, the PETAL trial, is currently exploring the therapeutic efficacy of combined TACE and pembrolizumab therapy [140,141]. Numerous studies are also underway evaluating the therapeutic benefits of combining LRT with various ICIs and systemic therapies, a topic that will be discussed in Section 10, Section 11 and Section 12.


**(C) TACE/COMBINATION OF CLASSES**


## 11. TACE/TKI/Immune Checkpoint Inhibitors

### 11.1. Technique and Periprocedural Management

As stated in Section 5, TACE is currently recommended for intermediate-stage HCC patients (BCLC B) and improves clinical efficacy before and also after curative resection [59]. To improve the efficacy of TACE procedures, newer trials are investigating the efficacy of combining TACE with both TKI and ICI. Sorafenib, the first-line systemic therapy for HCC, is a multi-kinase inhibitor that inhibits tumor angiogenesis and blocks the expression of VEGFR. Similarly, immune checkpoint inhibitors, such as nivolumab or pembrolizumab, inhibit PD-1, upregulating T-cell activation and proliferation and generating a more robust immune response against tumor cells [142]. Multiple medications were combined with TACE, and their efficacy is outlined in Table 2.

For patients receiving this combination, super-selective cTACE is generally conducted using imaging guidance; a catheter is positioned in the hepatic artery, followed by the placement of a super-selective microcatheter into the tumor’s direct blood supply. Administration of TKI and ICI, as well as their dosages and routes of administration, varied between trials, but generally, one cycle was started one month before or after TACE at a dose of around 3 mg/kg.

TKIs were administered orally twice a day, and ICIs were delivered daily via injection or in pill form [143]. The range in doses for ICIs varied from 8 mg to greater than 12 every 3 weeks and depended on patient weight [128,143,144]. Similarly, the exact dose administered to patients differed between studies and depended on additional factors such as their embolization condition, response to the medication, adverse effects, and the extent of their disease condition. Dose reductions were determined and recommended for those with side effects [128,144]. TACE was performed while systemic therapy was completed and was generally resumed 3–7 days afterward. Pre-administration labs, particularly those concerning transaminitis, were conducted prior to each round of systemic therapy [128,143].

### 11.2. Patient Selection

The efficacy of TACE declines with the number of TACE procedures. This diminishing response is termed “TACE failure” or “TACE refractory” and is used to describe scenarios where TACE can no longer control disease [144,145]. Kudo et al. characterize TACE-refractory patients as those exhibiting two or more consecutive ineffective responses in treated tumors—where viable lesions persist at 50% or more—even after changing chemotherapeutic agents. Additionally, patients may be defined as refractory if they experience two or more consecutive instances of disease progression in the liver despite changes in chemotherapy or if they show persistent tumor marker elevations, evidence of vascular invasion, or signs of extrahepatic spread following TACE treatments [145]. While patients with unresectable HCC (BCLC C) especially draw benefits from this combination therapy, patients excluded from studies using this combination therapy were as follows: patients with BCLC Class A or D, patients with Child–Pugh Class C, patients with a tumor size < 3 cm, patients with grade II–IV myelosuppression, signs of metastasis to the central nervous system, as well as patients with standard TACE contra-indications such as coagulopathy or metastatic spread [129,145,146].

### 11.3. Prognosis and Outcomes

Survival benefits linked to TAC/TKI/ICI are considerable. The included studies compared the survival benefits of late-stage HCC patients receiving combination therapy with TACE/TKI/ICI as compared to patients assigned to receive TACE combined with either ICI or TKI [129]. All included articles noted enhancement in both PFS and OS with the addition of ICIs to the TACE/TKI regimen [128,129,143,146]. The median OS also differed notably between the groups; for example, patients receiving the TACE/sorafenib group had a mean overall survival of 13.8 months, while this value was 23.3 months in the TACE/sorafenib/ICIs group [146]. Furthermore, the median PFS for patients receiving different ICI therapies was comparable. For instance, nivolumab OS was similar to that of pembrolizumab within the TACE/Sorafenib/ICI group at 13.6 months and 13.2 months, respectively [129,146]. The key findings of these studies are highlighted in Table 2.

Prognostic factors for PFS and OS in hepatocellular carcinoma treatments involving TACE with TKIs and ICIs include several key indicators. Factors such as Child–Pugh class, BCLC class, AFP levels, tumor size, metastasis, and the specific treatment regimen (TACE + Sor + ICI vs. TACE + Sor) were identified as significant predictors of PFS. Notably, BCLC Class C, high AFP levels, larger tumor size, and receiving TACE combined with both sorafenib and ICIs were associated with better PFS outcomes [129,146]. For OS, independent predictors included Child–Pugh class, BCLC class, the treatment regimen, and whether ablation was performed after disease progression, with each factor significantly influencing survival outcomes [129].

Regarding the technical application of these treatments, it was noted that the sequence of therapy application might impact efficacy. Administering TACE first potentially increases the efficacy of subsequent systemic therapies by reducing the tumor load and altering the microenvironment, which could enhance the uptake and effectiveness of sorafenib and ICIs [78,93,105]. This strategic approach also suggests a synergistic potential, maximizing tumor necrosis and minimizing progression rates. Further exploration into optimal sequencing and dosing could provide deeper insights into maximizing patient outcomes [78,128,143,147].

In general, adverse events seen in these patients are similar to those seen in patients who received TACE alone, such as transaminitis, post-ablation syndrome, and hepatic damage, among others [143,146,147]. The most common adverse effects were liver dysfunction or transaminitis, which was cited in all TACE/TKI/ICI studies; abdominal symptoms such as nausea, vomiting, or diarrhea, which were cited in all studies; and hand–foot syndrome, which were cited in three of the four articles included [128,129,143,146,147]. Table 2 notes other commonly noted complications. Moreover, while the combined therapy of TACE with TKI and ICIs shows promising results in terms of safety and efficacy, it also highlights the necessity for tailored therapeutic approaches based on individual patient characteristics, such as liver function and tumor staging, which could help in optimizing treatment outcomes and managing adverse effects [144,146]. 

## 12. TARE/Systemic Therapies

### 12.1. Technique and Periprocedural Management

There are no standardized guidelines for TARE/systemic therapy combinations, and only one prospective trial evaluates using this combination [101]. This area is among the newest in HCC management. Multiple clinical trials are actively ongoing or have recently ended. Of these trials, SORAMIC has some of the most robust data. In this trial, SIRT was performed after excluding relevant hepato-pulmonary shunts using 99mTc-labeled macroaggregated albumin (MAA) and ensuring there was no risk of microsphere misplacement in extrahepatic organs, as recommended by the manufacturer. SIRT with 90Y-resin microspheres was administered separately to each liver lobe, utilizing a selective segmental approach when appropriate. For patients with bilobar disease, SIRT was initially performed on the dominant diseased liver lobe, followed by treatment of the untreated contralateral lobe 4–6 weeks later. If the disease was confined to a single lobe, selective SIRT was performed in a single session. Dosing was individualized and adjusted patient-to-patient [101,148].

### 12.2. Patient Selection

SORAMIC was the only study to our knowledge that evaluated TARE with systemic therapies. Eligible patients were classified under BCLC stages A, B, and C, while all studies included patients with Child–Pugh scores ranging from A to B7. Inclusion allowed for those who had previously undergone resection or local/locoregional treatments. Importantly, patients who had received prior TACE or TAE were considered only if there was an interval and evidence of revascularization. This interval ranged from 4 weeks to 3 months. Additionally, patients with extrahepatic disease were included if they exhibited liver-dominant disease without pulmonary metastases [148]. Exclusion criteria focused on those with hepato-pulmonary shunts, previous external beam radiation therapy to the liver, or prior therapy with TKI or VEGF inhibitors. Furthermore, patients presenting with serum bilirubin levels exceeding 1.5 times the upper limit of the normal range were excluded to avoid complications and to ensure the reliability of study outcomes [101,148].

### 12.3. Prognostic Factors and Outcomes

Evaluation of radioembolization combined with systemic therapies is among the most anticipated trial data. There are three notable phase III trials that delineate the role of TARE in treating unresectable intermediate-to-advanced stage HCC: SARAH, SIRveNIB, and SORAMIC [148]. SARAH and SIRveNIB both compared the overall survival benefits of TARE with those of sorafenib [149,150]. On the other hand, SORAMIC evaluated the overall survival benefits of TARE + sorafenib with sorafenib alone [148]. The details and findings of these trials are outlined in Table 3. In SARAH and SIRveNIB trials, the median OS in the TARE arms ranged from 8 to 14 months, compared to 9.9 to 11.4 months in the sorafenib arms, and did not reach statistical significance. Similarly, in an overall study population, SORAMIC did not result in a significant improvement in overall survival compared with sorafenib alone [101,148]. The authors of these trials mentioned that this could be due to a multitude of factors: too broad of a patient selection protocol, older standard dosimetry, and variations in patient management across centers [101,148,149,150].

The SORAMIC trial defines grade 3–4 adverse events using the Common Terminology Criteria for Adverse Events (CTCAE). To elaborate, grade 3 events are those that are medically significant but not immediately life-threatening, and grade 4 events are those that are life-threatening [101,148,149,150]. The SORAMIC trial noted grade 3–4 events in 64.8% of patients in the TARE + sorafenib group, higher than the 53.8% seen in the sorafenib alone group or the 33.3% in patients who only received SIRT. Among the most common adverse events, hyperbilirubinemia was approximately three times more frequent in the SIRT plus sorafenib arm compared to the sorafenib arm (14.5% vs. 4.4%). Additionally, fatigue was significantly more prevalent in the SIRT plus sorafenib arm, affecting 35.2% of patients compared to 24.2% in the sorafenib arm [101,148,149,150]. Serious adverse events were more commonly reported in patients who received sorafenib (70.5%) than in patients who received SIRT + sorafenib (39.6%) [101,148]. Future studies examining TARE combined with systemic therapies could benefit from using patient populations more in line with those that draw benefits from sorafenib alone or TACE combined with TKIs—specifically, an ECOG performance status score of 0 or 1; Child–Pugh Class A.

## 13. Ablation/Systemic Therapies

### 13.1. Technique and Periprocedural Management

Five studies evaluated the benefits of RFA with systemic therapy. Patients underwent RFA on an inpatient basis. For details on the RFA technique, see Section 3. Patients then took oral therapy four days to one week after the first round of RFA. The dosage was similar to patients taking systemic therapy at 400 mg bid of sorafenib [151,152,153,154,155]. If patients had severe side effects or a severe deterioration of the quality of life (QoL) at the standard dosage, dose reductions or temporary interruptions were instituted [152]. For adverse drug reactions (ADRs) of grades 3–4, the dose was decreased to 200 mg twice daily until the ADRs improved to a grade of ≤2, then increased to 400 mg twice daily if well tolerated. Patients were treated with continuous sorafenib with no breaks before or after the new RFA procedure [151,153]. Further periprocedural management strategies mimicked those of standard systemic therapy procedures, outlined in Section 6.

### 13.2. Patient Selection

Patient selection for combination therapy with ablation and systemic therapy in HCC focused on different stages and sizes of HCC, including early small HCC, medium-sized HCC, and intermediate-stage HCC with a high tumor burden [151,153,154,155,156]. Generally, patients included were those with BCLC Stage 0–B1, BCLC Stage A, B, or C without extrahepatic disease, and those with Child–Pugh Class A liver function [152]. Exclusion criteria often encompassed severe coagulopathy, extensive prior treatment, or tumors located adjacent to large vessels or bile ducts to avoid complications during the ablation process. Additionally, patients who had not received previous systemic therapies and had relatively preserved liver function were preferred candidates for these combination treatments [153].

### 13.3. Prognosis and Outcomes

The prognosis and outcomes for HCC patients undergoing ablation therapies combined with systemic treatments are influenced by several key factors, and the major findings from studies evaluating this combination are presented in Table 4. The choice of combination therapy significantly affects patient outcomes. For example, combining sorafenib with RFA has been shown to improve recurrence rates and overall survival compared to RFA alone [154]. Studies such as those by Kan et al. and Feng et al. demonstrated that the addition of sorafenib led to a substantial decrease in recurrence rates and prolonged overall survival [153,154]. Additionally, lenvatinib combined with RFA has been shown to provide higher objective response rates and longer progression-free survival [155]. Among these combinations, lenvatinib plus RFA demonstrated the best outcomes, providing a better tumor response and improved survival rates compared to lenvatinib alone [152]. 

Patient selection criteria also play a crucial role in determining outcomes. Patients with early-stage HCC (BCLC stage 0 and A) and good liver function (Child–Pugh A or B) tend to have a better prognosis. The combination of systemic therapies like sorafenib and lenvatinib with RFA is particularly beneficial for these patients, as it enhances the efficacy of ablation by reducing tumor blood flow and increasing the extent of thermal coagulation [24,153]. Conversely, patients with advanced disease, significant comorbidities, or poor liver function may not experience the same benefits and could face increased risks from these combination therapies. This careful selection and timing of therapies are critical for optimizing patient outcomes. Future studies should evaluate how the selection, timing, and administration of systemic therapy, when combined with ablation, compare with ablation and systemic therapies alone [151,153,155].

## 14. Comparison and Future Directions

Our review highlights the promising potential of combining locoregional therapies such as TACE and ablation with systemic treatments, including targeted therapies and immune checkpoint inhibitors, in the management of HCC. The success of treatments for advanced HCC is gauged by overall survival, progression-free survival, time to progression, and response rates. Additional factors include rates of major adverse reactions such as bleeding, thrombosis, post-ablation syndrome, or others [24]. 

Selecting either LRT or systemic therapies involves a comprehensive assessment of liver function and tumor characteristics. Compared to systemic therapies, LRT can yield certain advantages and disadvantages for patients with advanced-stage HCC (BCLC C and D). For instance, even for patients outside of the Milan Criteria, LRT can serve as a bridge to curative therapies such as transplants [24,73]. Additionally, when clinically appropriate, LRTs can be used to provide symptomatic relief, improve quality of life, and reduce tumor burden [24,73]. The choice between LRTs also depends on the expertise and historical preferences of the treatment center. Systemic therapy is generally preferred for patients in later stages or those who cannot tolerate LRT-specific side effects. Although the most suitable patient populations, as well as adverse effects, are outlined in Section 3, Section 4, Section 5 and Section 6, the most notable side effects include PES, bleeding, bile duct injuries, and hepatic decompensation. In patients with multilobar disease, systemic therapy is generally recommended as it targets not only the primary liver tumor but also any metastatic sites, offering a broader treatment scope that is essential for advanced disease with extrahepatic spread [24,145].

Our study noted the multiple advantages of combining LRTs with systemic therapies. Notably, key findings of LRT + systemic therapy combinations are illustrated in Table 2. Among the various strategies, RFA plus lenvatinib and TACE plus sorafenib emerged as the combinations with the best outcomes for specific patient groups. For early-stage HCC patients (BCLC stage 0 and A), RFA plus lenvatinib demonstrated superior outcomes with longer progression-free survival and higher objective response rates. On the other hand, for intermediate-stage HCC patients, TACE with sorafenib showed the best results in prolonging overall survival and reducing recurrence rates. Although there was only one study that we identified focused on TARE combined with systemic therapy, this trial showed no benefit for overall survival or progression-free survival as compared to systemic therapy alone, highlighting the need for future research on this specific combination. The variability in both patient populations and treatment strategies suggests that there is a need for tailored approaches based on the disease stage to optimize patient outcomes.

However, these combination therapies are not without adverse events. The TACTICS trial for TACE plus sorafenib reported significant adverse effects, including elevated ALT/AST levels (23.4% and 22.1%, respectively), thrombocytopenia (13.0%), and hypertension (10.4%) [126,127]. Similarly, combining RFA with lenvatinib led to high rates of adverse events such as ALT elevation (88.9%), hypertension (44.4%), and fatigue (55.6%) [155]. Among the combinations, TACE plus lenvatinib had some of the most severe adverse events, including abdominal pain (50.6%), fever (88.8%), and hand–foot skin reaction (31.2%), among others, indicating the need for careful management and monitoring of patients undergoing these treatments. The synergistic potential of combination therapies should be fully utilized for patients who can tolerate such treatments. However, several crucial questions remain unanswered. How should we identify the ideal candidates for combination therapies? At what point should patients transition between different treatment modalities? How can we best preserve major organ functions to facilitate uninterrupted treatment? The safety, feasibility, and efficacy of combined systemic and locoregional therapies must be validated through ongoing research to improve outcomes and quality of life for patients with advanced HCC.

While locoregional and systemic therapies, both individually and in combination, offer unique benefits in controlling tumor progression and enhancing prognosis, the long-term outlook remains challenging. This is partially because of the difficulty in standardizing patients: multiple trials experienced challenges and noted a lack of benefit in treatment because of poor standardization among patients, poor adherence to treatment, and loss of patient follow-up. Multiple studies also noted that BCLC B or C would benefit from treatment the most, yet included those who were staged as BCLC A, potentially skewing results [101,147,148,152]. Furthermore, the heterogeneity in study designs, patient populations, and treatment protocols poses significant challenges in directly comparing outcomes across different trials. Variations in inclusion and exclusion criteria, such as differences in liver function status, tumor burden, and prior treatments, can lead to inconsistencies in reported efficacy and safety profiles. Additionally, the lack of standardized dosing regimens and timing for the administration of systemic therapies in combination with locoregional treatments further complicates the interpretation of results and the formulation of universally applicable treatment guidelines. The efficacy of different combination therapies could be evaluated in a clearer manner if studies used standardized measures and time points for overall survival, TTP, and PFS. In regard to systemic therapy/TACE combinations, multiple critical questions persist, particularly pertaining to the identification of predictive factors (such as treatment-related adverse events, PD-L1 expression, or tumor mutational burden) to effectively select patients suitable for combined locoregional therapies and immunotherapy. Addressing these uncertainties not only holds crucial implications for disease monitoring but also for optimizing treatment decisions. Thus, these combined approaches represent a promising avenue for future research in this field [124,126,129].

Another limitation is the potential for selection bias in clinical trials. Patients who are eligible and willing to participate in clinical studies may not represent the broader HCC population, particularly those with more advanced diseases or comorbidities that preclude trial participation. This could result in an overestimation of treatment benefits and an underestimation of adverse events. Moreover, the relatively short follow-up periods in some studies, such as the SARAH trial, may not fully capture long-term outcomes and late-onset adverse effects, which are crucial for assessing the durability and safety of combination therapies [87,150]. Cultural factors may also play a significant role in the implementation and outcomes of these treatments, according to the studies. Differences in healthcare infrastructure, patient preferences, and the availability of specific therapies across various regions can impact treatment adherence and effectiveness. Future research should aim to address these limitations by standardizing trial designs, identifying the most relevant patient populations, harmonizing treatment protocols, and extending follow-up durations.

## 15. Conclusions

This review analyzes the evolving landscape of HCC treatment, particularly highlighting the integration of locoregional therapies with systemic treatments. Along with denoting the techniques, periprocedural management, and patient selection behind these strategies, the findings suggest that combining LRTs with systemic therapies offers significant improvements in PFS, OS, and response rates compared to locoregional or systemic therapies alone. Similarly, the heterogeneity in study designs and patient populations underscores the importance of tailoring combination therapies based on disease stage and patient characteristics to optimize efficacy. However, this analysis does not come without limitations, including difficulty in analyzing studies due to differences in study designs, patient populations, and potential selection biases. Future studies should aim to standardize treatment protocols, extend follow-up durations, and incorporate standardized markers for success to ensure broader applicability and better patient outcomes. Continued research and clinical trials will be essential in refining these approaches, ensuring they provide maximum benefit with manageable adverse effects for diverse patient populations.

## Figures and Tables

**Figure 1 biomedicines-12-01432-f001:**
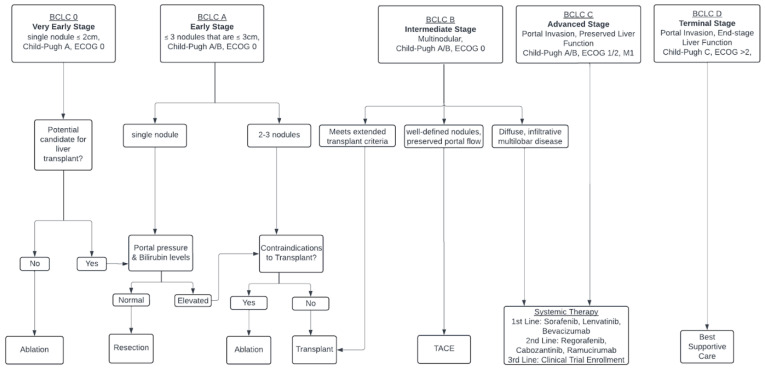
Barcelona Guidelines for Hepatocellular Carcinoma Treatment.

**Table 1 biomedicines-12-01432-t001:** Comparison of Systemic Treatments for Advanced Hepatocellular Carcinoma. Adapted from [102].

Type of Medication	Systemic Therapies	Line of Therapy	Benefits	Risks/Complications
Multikinase Inhibitor	Sorafenib	First Line	Prolongs OS	Hand–foot skin reaction, fatigue, hypertension, gastrointestinal issues
	Lenvatinib	First Line	Comparable OS to sorafenib, Higher objective response rate (ORR), Longer progression-free survival (PFS) and time-to-progression (TTP)	Hypertension, proteinuria, fatigue, hypothyroidism
	Regorafenib	Second Line	Prolongs OS; Improves PFS and TTP; Higher disease control rate (DCR)	Hand–foot skin reaction, hypertension, diarrhea, transaminitis
	Cabozantinib	Second Line	Prolongs OS and PFS; Improves ORR and DCR	Hypertension, palmar-plantar erythrodysesthesia, diarrhea, fatigue
Monoclonal Antibody	Ramucirumab	Second Line	Prolongs OS and PFS in patients with high AFP	Hypertension, hyponatremia, increased AST
Immune Checkpoint Inhibitor (PD-1)	Pembrolizumab	Second Line	Prolongs OS and PFS; Improves ORR	Rash, diarrhea, pruritus, immune-related adverse events
	Nivolumab	Second Line	Prolongs OS; Improves ORR	Rash, diarrhea, pruritus, immune-related adverse events
Immune Checkpoint Inhibitors (CTLA-4)	Ipilimumab	Second Line	Significant OS benefit; High ORR	Immune-mediated toxicities
VEGF Inhibitor	Bevacizumab	First Line	Prolongs OS; Higher ORR and complete response rate (CRR); Better HRQOL	Risk of bleeding, proteinuria, autoimmune events

**Table 2 biomedicines-12-01432-t002:** Summary of trials investigating sorafenib/TACE combination therapy as well as their treatment administration and technique.

Key Features	*POST-TACE* [130]	*TACE-2* [121]	*SPACE* [131]	*TACTICS* [123,124]
Study Design	Phase 3 Trial	Open-label, randomized, investigator-initiated, phase 3 trial	Multicenter, phase II randomized, double-blind, placebo-controlled study	Multicenter, randomized, open-label, multicenter prospective trial
Objective	Evaluate sorafenib in Japanese and Korean patients with unresectable HCC responsive to TACE	Compare TACE + sorafenib vs. TACE + placebo for progression-free survival	Evaluate DEB-TACE + sorafenib in intermediate stage HCC	Compare TACE + sorafenib vs. TACE alone using a TACE-specific endpoint with pre-treatment of sorafenib
Inclusion Criteria	Aged ≥ 18 years of age with unresectable HCC; Child–Pugh A cirrhosis; Eastern Cooperative Oncology Group (ECOG) performance status (PS) 0 or 1; Adequate bone marrow, liver, and renal function	Aged ≥ 18 years of age with histological or non-invasive HCC; Child–Pugh A liver disease; ECOG PS ≤ 1; Adequate bone marrow, liver, and renal function	Aged ≥ 18 years with unresectable, multinodular, asymptomatic HCC (BCLC stage B); Child–Pugh Class A; ECOG PS of 0; Adequate bone marrow, liver, and renal function	Aged ≥ 20 years with HCC Child–Pugh score ≤ 7; ECOG PS 0 or 1; Preserved organ function.
Exclusion Criteria	Macroscopic vascular invasion or extrahepatic invasion; prior systemic agents for advanced HCC; chronic comorbidities such as renal failure, cardiac disease history, or active serious infection	Extrahepatic metastasis; previous embolization, systemic therapy, or radiotherapy for HCC	HCC with vascular invasion, extrahepatic tumor spread, or advanced liver disease (Child–Pugh Class B or C); previous local treatment (resection, RFA, PEI, or TACE); gastrointestinal bleeding, encephalopathy, or ascites)	HCC with diffuse tumor lesions or extrahepatic metastases; previous treatment for advanced HCC, including systemic chemotherapy; chronic comorbidities such as cardiac disease, active serious infections, hepatic encephalopathy, or uncontrolled ascites
Sequence of Administration	Sorafenib was administered after TACE; the trial only included responders to TACE, and 60% of patients started drug treatment 9 weeks after TACE	Sorafenib was administered first, followed by DEB-TACE 2–5 weeks later	Treatment was administered in 4-week cycles; sorafenib started on day 1, and the first DEB-TACE session followed 3–7 days later; subsequent TACE sessions were scheduled around days 1 of cycles 3, 7, and 13 (±4 days)	400 mg of oral sorafenib was given 2–3 weeks prior to the initial TACE; sorafenib was paused 2 days before TACE and restarted 3 days post-sessions, with the dose potentially increased to 800 mg based on investigator discretion. Repeat TACE was advised if viable lesions exceeded 50% of baseline tumor volume before the first session
Systemic Therapy (Sorafenib) Administration	400 mg of oral sorafenib once daily, continuously	400 mg of oral sorafenib twice daily, continuously	400 mg of oral sorafenib twice daily, continuously	400 mg of oral sorafenib daily for 2–3 weeks before TACE, followed by 800 mg daily during on-demand conventional TACE sessions
TACE Administration	TACE was performed using gelatin foam and lipiodol, along with chemotherapeutic agents, including epirubicin, cisplatin, doxorubicin, and mitomycin	DEB-TACE was administered 2–5 weeks after randomization using doxorubicin-loaded drug-eluting beads (150 mg) and delivered via the hepatic artery accessed through the femoral artery with a super-selective approach	DEB-TACE with doxorubicin-loaded drug-eluting beads (150 mg) was administered 3–7 days after starting sorafenib; subsequent TACE sessions occurred around days 1 of cycles 3, 7, and 13 (± 4 days)	TACE was performed using lipiodol mixed with either epirubicin or miriplatin, followed by embolization using an embolic agent (Gelpart) to block the tumor-feeding artery
Overall Survival (OS)	29.7 months (sorafenib) vs. not reached (placebo)	21.1 months (sorafenib) vs. 19.7 months (placebo)	18.2 months (sorafenib) vs. 18.4 months (placebo)	36.2 months (combined therapy) vs. 30.8 months (TACE alone)
Progression Free Survival (PFS)	5.4 months (sorafenib) vs. 3.7 months (placebo)	7.8 months (sorafenib) vs. 7.7 months (placebo)	NDA	22.8 (combined therapy) vs. 13.5 months (TACE alone)
Median Time to Progression (mTTP)	5.4 months (sorafenib) vs. 3.7 months (placebo)	10.7 months (sorafenib) vs. 10.5 months (placebo)	5.5 months (sorafenib) vs. 5.4 months (placebo)	NDA

**Table 3 biomedicines-12-01432-t003:** Trials investigating TARE combination therapy.

Key Features	SARAH [147]	SIRveNIB [146]	SORAMIC [145]
Study Design	Multicenter randomized controlled phase 3 trial	Open-label, randomized, investigator-initiated, phase 3 trial	Multicenter randomized controlled phase 2 trial
Objective	Compare efficacy and safety of SIRT with Y-90 resin microspheres vs. sorafenib in patients with locally advanced/inoperable HCC	Compare efficacy and safety of Y-90 resin microspheres RE vs. sorafenib in patients with locally advanced HCC	Compare the efficacy of TACE combined with sorafenib vs. TACE alone in patients with advanced HCC
Inclusion Criteria	Aged ≥ 18 years; life expectancy > 3 months; ECOG 0 or 1; Child–Pugh A or B7; BCLC stage C; not eligible for resection, liver transplantation, or thermal ablation; HCC with two unsuccessful TACE rounds	Aged ≥ 18 years; BCLC stage B or C without extrahepatic disease; with/without PVT; ≤2 previous hepatic artery-directed therapies; no hepatic artery-directed treatment within last 4 weeks	Unresectable HCC; aged ≥ 18 years; ECOG ≤ 2; adequate liver function (Child–Pugh A/B); no prior systemic therapy
Exclusion Criteria	Another primary tumor; extrahepatic metastasis; previous treatment of the current nodule; active GI bleeding/encephalopathy/refractory ascites; contraindications to hepatic embolization	More than two previous hepatic artery-directed therapies; prior sorafenib or VEGF inhibitors; prior radiotherapy	Previous systemic therapy for HCC; significant comorbidities affecting liver function; extensive prior liver resection
Overall Survival (OS)	Median OS: 8.0 months (SIRT) vs. 9.9 months (sorafenib)	Median OS: 8.8 months (RE) vs. 10.0 months (sorafenib)	TARE + sorafenib: 12.1 months vs. sorafenib alone: 11.4 months (ITT); TARE + sorafenib: 14 months vs. Sorafenib alone: 11.1 months (PP)
Adverse Events	Serious AEs: 77% (SIRT) vs. 82% (sorafenib)	Serious AEs: 20.8% (RE) vs. 35.2% (sorafenib)	Higher AE rates with sorafenib; lower rates with combined therapy
Quality of Life	Better in the SIRT group than in the sorafenib group	Improved toxicity profile in the RE group	Improved quality of life in a combination therapy group

**Table 4 biomedicines-12-01432-t004:** Evaluating the efficacy and adverse events for combination therapies.

Modality	Patient Selection	Median Overall Survival (mos)	Time to Progression	Median Progression Free Survival (mos)	Response Rates (%)	Risks and Complications (Grade 3)
Systemic Therapies						
Sorafenib [110,154] *SHARP, Asia–Pacific*	Unresectable HCC, ECOG 0, or 1, Child–Pugh Class A liver function	6.5, 10.7	2.8, 5.5	NDA	2%, 3.3%	Hand–foot skin reaction (8%), diarrhea (8%), weight loss (2%)
Atezolizumab + Bevacizumab [119,155,157] *IMbrave150*	Unresectable HCC, ECOG 0 or 1, Child–Pugh Class A liver function	19.2	6.8	6.9	27.3%	Proteinuria (29%), hypertension (28%), increased aspartate aminotransferase (16%), fatigue (16%)
Durvalumab + Tremelimumab [120,156,157,158,159,160] *HIMALAYA*	Unresectable HCC, ECOG 0 or 1, Child–Pugh Class A liver function	18.7	5.4	2.17	20.1%	Increased aspartate transaminase (5.2%), lipase increased (6.2%), hypertension (1.8%)
TACE + Systemic Therapy						
TACE + Sorafenib [123,124] *TACTICS*	Unresectable HCC, ECOG 0 or 1, Child–Pugh Class A ≤ 7	36.2	NDA	25.2 (TACE-specific)	71.3%	Elevated ALT/AST (23.4% and 22.1%, respectively), thrombocytopenia (13.0%), hypertension (10.4%), elevated lipase (14.3%)
TACE + Lenvatinib [134] *LAUNCH*	Advanced primary HCC without prior treatment or initial, recurrent advanced HCC post-radical resection without postoperative treatment, ECOG 0 or 1, Child–Pugh Class A	17.8	NDA	10.6	54.1% (modified RECIST)	Abdominal pain (50.6%), fever (38.8%), nausea (35.9%), hand–foot skin reaction (31.2%), elevated ALT/AST (21.2% and 26.5%, respectively), hyperbilirubinemia (17.9%), hypoalbuminemia (14.3%)
TACE + Immune Checkpoint Inhibitors (ICI)						
TACE + Nivolumab [135,136] *IMMUTACE*	Intermediate-stage HCC, ECOG 0 to 2, Child–Pugh score of A	28.3	NDA	7.2	71.4% (modified RECIST)	Fatigue (30.6%), increased AST/ALT (24.5% and 22.4%, respectively)
TACE + Combination of Systemic/ICI						
TACE + Sorafenib + nivolumab/pembrolizumab [125]	BCLC C, ECOG PS 0 or 1, Child–Pugh Class A/B	23.3	NDA	13.6 (nivolumab) 13.2 (pembrolizumab)	41.7%	Transaminitis (45.8%), fever (58.3%), gastrointestinal reaction (41.7%), hand–foot syndrome (37.5%)
TACE + TKI + PD-1i [144]	BCLC C, ECOG PS 0 or 1, Child–Pugh Class A/B	16.9	NDA	7.3 (not statistically significant)	56.1%	Fever (58.3%), liver dysfunction (50%), nausea and vomiting (20.8%), hand–foot syndrome (41.7%), thyroid dysfunction (16.7%)
TACE + Sorafenib + Camrelizumab TACE + Lenvatinib + Sintilimab [78]	BCLC C, ECOG PS 0 or 1, Child–Pugh Class A/B	12.9	4.9	4.9	17%	Transaminitis (7%), fatigue (17%), diarrhea (11%), adrenal insufficiency (2%)
TACE + Sorafinib + sintilimab/camrelizumab [143]	BCLC C, Child–Pugh Class A/B	21.63	NDA	7.63	59.09%	Thyroid dysfunction (43%), hepatocyte dysfunction (100%), nausea (45.5%), vomiting (27.3%), hand–foot syndrome (16%)
TARE + Systemic Therapy						
TARE + Sorafenib [145,146,147]	BCLC A, B, C; Child–Pugh Class A-B7	12.1 (not statistically significant)	NDA	NDA	NDA	Grade 3–4 adverse events (64.8%), hyperbilirubinemia (14.5%), fatigue (35.2%), hand–foot syndrome (9.4%)
Ablation + Systemic Therapy						
RFA + Sorafenib [148]	BCLC A, Child–Pugh Class A,B (Kan)	NDA	17.0 (Fukuda) 12.3 (Gong)	NDA	NDA	Hand–foot reaction (83.3%), diarrhea (46.7%)
RFA + Lenvatinib [153]	BCLC B2, Child–Pugh A	21.3 months	12.5 (wang)	NDA	100%	Transaminitis (88.9%), hypertension (44.4%)

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
