# Peer review of "Examining the Efficacy and Safety of Combined Locoregional Therapy and Immunotherapy in Treating Hepatocellular Carcinoma"

_biomedicines, 2024, doi:10.3390/biomedicines12071432_

Round 1
Reviewer 1 Report
Comments and Suggestions for Authors
The manuscript focused on the current status of liver cancer treatment, including efficacy and safety of combined locoregional therapy and immunotherapy for hepatocellular carcinoma. In general, this review is comprehensive and reasonable. So this manuscript may be suitable for publication in Biomedicines after minor revision.
1. The format of Table 1, Table 2 and Table 3 is different.
2. The description of TKI treatment in Chapter 7 is limited and needs to be discussed more.
3. Compared with systemic therapies, what are the advantages and disadvantages of LRT in the advanced treatment of liver cancer patients, and how to choose a more appropriate treatment in clinical treatment?
Author Response
On behalf of the authors, I am sending you a corrected copy of the paper entitled “Examining the Efficacy and Safety of Combined Locoregional Therapy and Immunotherapy in Treating Hepatocellular Carcinoma” submitted on May 28, 2024. Changes are highlighted in yellow in attachment. Similarly, the suggestions made are addressed as follows below, with comments underlined and responses underneath:
- The description of TKI treatment in Chapter 7 is limited and needs to be discussed more.
- More information has been added to this section. Key indications for systemic therapies, specifically TKIs is expanded upon. Similarly, the benefits and risks tied with TKIs that are outlined in Table 1, along with new information regarding the most common adverse effects, was added in section 7.3.
- New information regarding the mechanism of action for TKIs is expanded upon in section 7.1.
- The format of Table 1, Table 2 and Table 3 is different.
- Thank you for the feedback. Table 1 is oriented towards pointing out the respective systemic therapies mechanism of action, line of therapy, benefits, and risks. The breath of research varies between systemic therapies and combination therapies, with systemic therapies being much more established. For this reason, we have combined Tables 2 and 3, with this table focused on relaying key information from landmark clinical trials as well as place increased focus on technique and treatment delivery.
- Compared with systemic therapies, what are the advantages and disadvantages of LRT in the advanced treatment of liver cancer patients, and how to choose a more appropriate treatment in clinical treatment?
- Thank you for mentioning this point. A new paragraph is added in the discussion discussing this. This paragraph focuses on the main considerations when clinically choosing between LRTs alone and systemic therapies.

Reviewer 2 Report
Comments and Suggestions for Authors The presented review article analyzes the evolving landscape of HCC treatment, particularly highlighting the integration of locoregional therapies with systemic treatments. The presented article is an excellent review work. I only have two comments about the layout. (1) Table 5. In this table, the spaces between entries are too large, so the table has become too large. It takes up four pages and looks ugly. In my opinion, this table should fit on two pages with reduced spaces between entries. (2) References. All reference list entries must be carefully reviewed. In the current version, the entries do not correspond to the journal style. The punctuation marks between authors' surnames and first names need to be changed. All journal names must be presented in italics. Publication years and issues must be presented in the appropriate format.Author Response
On behalf of the authors, I am sending you a corrected copy of the paper entitled “Examining the Efficacy and Safety of Combined Locoregional Therapy and Immunotherapy in Treating Hepatocellular Carcinoma” submitted on May 28, 2024. Changes are highlighted in yellow in attachment. Similarly, the suggestions made are addressed as follows below, with comments underlined and responses underneath:
- Table 5. In this table, the spaces between entries are too large, so the table has become too large. It takes up four pages and looks ugly. In my opinion, this table should fit on two pages with reduced spaces between entries.
- Table 5, now labeled Table 4, is 2.5 pages. This was the most that it could be shrunk!
- References. All reference list entries must be carefully reviewed. In the current version, the entries do not correspond to the journal style. The punctuation marks between authors' surnames and first names need to be changed. All journal names must be presented in italics. Publication years and issues must be presented in the appropriate format.
- Thank you for this comment! The references have been adjusted according to journal guidelines and reviewer feedback.
